# Reduced Type 2 Innate Lymphocyte Cell Frequencies in Patent *Wuchereria bancrofti*-Infected Individuals

**DOI:** 10.3390/pathogens12050665

**Published:** 2023-04-30

**Authors:** Ruth S. E. Tamadaho, Jubin Osei-Mensah, Kathrin Arndts, Linda Batsa Debrah, Alexander Y. Debrah, Laura E. Layland, Achim Hoerauf, Kenneth Pfarr, Manuel Ritter

**Affiliations:** 1Institute for Medical Microbiology, Immunology and Parasitology (IMMIP), University Hospital Bonn (UKB), 53127 Bonn, Germany; 2Kumasi Centre for Collaborative Research in Tropical Medicine (KCCR), UPO, PMB, Kumasi 00233, Ghana; 3Department of Pathobiology, School of Veterinary Medicine, Kwame Nkrumah University of Science and Technology, UPO, PMB, Kumasi 00233, Ghana; 4Department of Clinical Microbiology, School of Medicine and Dentistry, Kwame Nkrumah University of Science and Technology, UPO, PMB, Kumasi 00233, Ghana; 5Faculty of Allied Health Sciences, Kwame Nkrumah University of Science and Technology, UPO, PMB, Kumasi 00233, Ghana; 6German Center for Infection Research (DZIF), Partner Site Bonn-Cologne, 53127 Bonn, Germany; 7German-West African Centre for Global Health and Pandemic Prevention (G-WAC), Partner Site Bonn, 53127 Bonn, Germany

**Keywords:** *Wuchereria bancrofti*, innate lymphoid cells (ILCs), myeloid-derived suppressor cells (MDSCs), M2 macrophages, mass drug administration, cured infection, filarial-specific immune responses, immune modulation

## Abstract

Approximately 51 million individuals suffer from lymphatic filariasis (LF) caused mainly by the filarial worm *Wuchereria bancrofti*. Mass drug administration (MDA) programs led to a significant reduction in the number of infected individuals, but the consequences of the treatment and clearance of infection in regard to host immunity remain uncertain. Thus, this study investigates the composition of myeloid-derived suppressor cells (MDSCs), macrophage subsets and innate lymphoid cells (ILCs), in patent (circulating filarial antigen (CFA)+ microfilariae (MF)+) and latent (CFA+MF−) *W. bancrofti*-infected individuals, previously *W. bancrofti*-infected (PI) individuals cured of the infection due to MDA, uninfected controls (endemic normal (EN)) and individuals who suffer from lymphoedema (LE) from the Western Region of Ghana. Frequencies of ILC2 were significantly reduced in *W. bancrofti*-infected individuals, while the frequencies of MDSCs, M2 macrophages, ILC1 and ILC3 were comparable between the cohorts. Importantly, clearance of infection due to MDA restored the ILC2 frequencies, suggesting that ILC2 subsets might migrate to the site of infection within the lymphatic tissue. In general, the immune cell composition in individuals who cured the infection were comparable to the uninfected individuals, showing that filarial-driven changes of the immune responses require an active infection and are not maintained upon the clearance of the infection.

## 1. Introduction

Data from 2018 revealed that around 51 million individuals are still infected with lymphatic filariasis (LF) [1]. LF is a mosquito-borne filarial infection caused by worm parasites such as *Wuchereria bancrofti*, *Brugia malayi* or *Brugia timori*. However, *W. bancrofti* is responsible for over 90% of the infections worldwide, especially in Africa [2]. Mature female worms release microfilariae (MF, the offspring of the worm) that can be detected in whole blood as well as circulating filarial antigen (CFA), and both are used to ascertain *W. bancrofti* infection [2,3,4]. In LF endemic areas, five groups of individuals can be differentiated as follows: endemic normals (EN) that encompasses individuals who have lived at least five years in a community where they are exposed to the infection but remain uninfected, patent (CFA+MF+) and latent (CFA+MF−) *W. bancrofti*-infected individuals, as well as individuals who suffer from lymphoedema (LE) of which the majority have cleared the infection or harbor only few MF, and previously infected (PI) individuals who had a patent or latent infection but became CFA−MF− due to participation in the ivermectin (IVM) mass drug administration (MDA) program [5].

In general, it was shown that filarial infections increase the frequencies of peripheral immune cell populations including T and B cells, monocytes and granulocytes [6,7]. In detail, the immunological characterizations of *W. bancrofti*-infected, LE and EN individuals show that *W. bancrofti* (e.g., CFA+MF+ individuals) induce increased levels of regulatory T (Treg) and B cells (Breg) in the peripheral blood accompanied by regulatory and type 2 immune responses including IL-10, IL-4, IL-5, TGF-β and IgG4 [5,8,9,10,11,12]. In contrast, LE individuals are associated with enhanced Th1, Th17 and pro-inflammatory responses [12,13,14,15,16], and harbor exhausted T cell populations in peripheral blood [17,18]. However, reports about the immunological characterization of individuals who cleared the infection (PI; previously infected individuals) due to MDA are sparse. Nevertheless, we have previously shown that regulatory immune cells in the periphery are reduced in PI compared to *W. bancrofti*-infected individuals [5], suggesting that the immune profile of the host changes after the clearance of the filarial infection.

Although macrophages, innate lymphoid cells (ILCs) and myeloid-derived suppressor cells (MDSCs) were shown to play a crucial role during inflammation as well as helminth infections [19,20,21,22,23,24], their role during *W. bancrofti* infection remains uncertain. Indeed, macrophages can express markers such as CD206, CD200R and CD163 [25]. Treatment with migration inhibitory factor proteins (found on the filarial parasites *W. bancrofti* and *B. malayi*) elicited M2-associated markers on murine macrophages in vitro, but in vivo alternative activation was also demonstrated in various helminth models [26,27,28,29,30]. With regard to ILC2 expansion in individuals infected with helminths, research on *Schistosoma haematobium* infection found lower frequencies of these cells in the peripheral blood of the egg-positive infants when compared to the egg-negative group [31]. In contrast, another study showed a high percentage of ILC2s in a cohort of mixed filarial-infected people with very small group sizes [32]. A study using the *Litomosoides sigmodontis* mouse model demonstrated an increase in ILC2s in the pleural cavity of C57BL/6 mice compared to BALB/c mice and reported that ILC2 depletion using anti-CD90.2 antibodies could augment the release of microfilariae in Rag2-deficient C57BL/6 mice. In addition, ILC2 numbers were positively correlated with worm burden [33], indicating that ILC2 are an important cell population for the filarial-driven modulation of host immunity. In addition, we have recently shown that monocytic (Mo-MDSCs) and polymorphonuclear MDSCs (PMN-MDSCs) accumulate at the site of infection (pleural cavity) in BALB/c mice and their levels positively correlated with worm burden. Interestingly, only Mo-MDSCs could suppress the production of IFN-γ and IL-13, but not IL-5, by *L. sigmodontis*-specific CD4+ T cells [23]. Findings from animal models may not be directly translated to human filarial infections and information about ILC, MDSC and M2 macrophage populations during *W. bancrofti* infection is scarce. Thus, we investigated the composition of these immune cells in peripheral blood from individuals living in a *W. bancrofti* endemic area in the Western Region of Ghana, where IVM MDA programs were implemented.

## 2. Materials and Methods

### 2.1. Ethics Statement

A case–control study was conducted in 2009 involving 1774 Ghanaian volunteers within the health districts Nzema East and Ahanta West of the Western Region of Ghana to identify genetic biomarkers which are associated with different manifestations of LF [34,35]. The current study is part of a follow-up study that took place in 2015 conducted with the approval of the Committee on Human Research, Publications and Ethics at the School of Medical Sciences of the Kwame Nkrumah University of Science and Technology (KNUST), the Komfo Anokye Teaching Hospital, Kumasi, Ghana (CHRPE/AP/026/12; CHRPE/AP/112/13; CHRPE/AP/005/14; CHRPE/AP/018/15), and the Ethics Committee of the University Hospital of Bonn, Germany (018/12). The health directorates of the Nzema East and Ahanta West districts in Ghana also provided authorization for this study. Procedures and the purpose of the study were explained, and verbal and written informed consent was obtained from community leaders and all participants, respectively.

### 2.2. Sample Collection

Peripheral blood was obtained from uninfected endemic normals (EN), lymphoedema (LE) patients, patent (CFA+MF+) and latent (CFA+MF−) *W. bancrofti*-infected individuals as well as individuals who were previously infected with *W. bancrofti* (PI) and cleared the infection. To determine the presence of MF, night blood (due to nocturnal periodicity) [36] was obtained from the participants and analyzed using thick blood film and the Sedgewick Rafter counting technique as described [5]. In addition, circulating filarial antigen (CFA) specific for *W. bancrofti* was detected using immunochromatographic card tests (ICT) from the BinaxNOW^®^ Filariasis kit (Alere, Cologne, Germany) according to the manufacture’s protocol. LE individuals were characterized based on the presence of oedema on the upper and lower limb extremities as described by Dreyer and colleagues [37]. Participants were also tested for *Plasmodium* and *Mansonella* infections using blood smear analysis, but neither infection was detected. Of note, *Onchocerca volvulus* is absent at the study site.

### 2.3. Cryopreservation of Whole Blood Cells

From each individual, 100 µL of blood was treated with red blood lysis buffer (Biolegend, San Diego, CA, USA) according to manufacturer’s instruction to eliminate red blood cells. Red blood cell-depleted immune cells were then fixed and permeabilized using a Fixation/Permeabilization kit (Thermo Fisher Scientific, Grand Island, NE, USA) according to the manufacturer’s protocol. After permeabilization, cells were washed and placed in 200–250 µL freezing medium containing 90% FCS and 10% dimethyl sulfoxide and stored at −80 °C. Frozen samples were then shipped in liquid nitrogen to Germany for flow cytometry analysis.

### 2.4. Flow Cytometry Staining of MDSCs, ILCs and M2 Macrophages

To assess the composition and function of MDSCs, ILCs, and M2 macrophages, multi-color flow cytometry panels were established. Cryopreserved cells were thawed and washed twice with permeabilization buffer (Thermo Fisher Scientific). Thereafter, cells were stained with a combination of fluorophore-conjugated anti-human monoclonal antibodies. In detail, antibodies for the discrimination of MDSCs, M2 macrophages and ILCs included anti-HLA-DR-BUV661 (clone G46-6), anti-CD33-Super Bright 436 (clone P67.6), anti-CD11b-FITC (clone ICRF 44), anti-IL-10-PE (clone JES3-9D7), anti-CD15-PerCP e-Fluor 710 (clone MMA), anti-CD14-PE-Cyanine 7 (clone 61D3), anti-IL-6-APC (clone MQ2-13A5), anti-HLA-DR- BUV661 (clone G46-6), anti-CD206-eFluor 450 (clone 19.2), anti-IL-10-PE (clone JES3-9D7), anti-CD163-Super Bright 600 (clone eBioGHI/61 (GHI/61), anti-Lineage-FITC (see details on Lineage antibodies in Appendix A), anti-CD161-PE-eFluor 610 (clone HP-3G10), anti-CD200R-PerCP-eFluor 710 (clone OX108), anti-IL-4-PE-Cyanine 7 (clone 8D4-8), anti-CD294-APC (clone BM16), anti-CD127-APC-eFluor 780 (clone eBioRDR5) and anti-CD117-BUV395 (clone 104D2) antibodies. All antibodies were purchased from Thermo Fisher Scientific, except anti-CD117-BUV395 and anti-HLA-DR-BUV661 that were from BD Biosciences, Heidelberg, Germany. Cells were acquired using the CytoFlex S flow cytometer (Beckman Coulter, Brea, CA, USA) and analyzed using the FlowJo_v10.6.0 software (FlowJo LLC, Ashland, OR, USA). Gating strategies were developed based on fluorescence-minus-one controls, and compensation was performed using VersaComp Antibody capture kit (Beckman Coulter). The gating strategies for MDSCs [38,39,40], M2 macrophages [41] and ILCs [42,43] were based on previous publications. All figures show frequencies of total cells (“all cell gate” within Appendix A).

### 2.5. Statistical Analysis

Statistical analyses and graphs were generated using the GraphPad Prism 7.05 (GraphPad Software, Inc., La Jolla, San Diego, CA, USA). The non-parametric data were tested using a Kruskal–Wallis test with the Dunn’s post hoc test for comparison of the groups. For correlation analysis, statistical significances were tested using the Spearman correlation test. *p*-values of ≤0.05 were considered significant.

## 3. Results

### 3.1. Study Population

To investigate the composition of MDSCs, M2 macrophages and ILCs, cryopreserved peripheral blood cells were analyzed from the health districts Nzema East and Ahanta West of the Western Region of Ghana, an area in which MDA treatment (400 mg albendazole + 200 µg/kg ivermectin) was performed. In total, cryopreserved cells from 89 EN, 54 CFA+MF− (latent *W. bancrofti* infection), 14 CFA+MF+ (patent *W. bancrofti* infection), 85 LE and 117 PI individuals were investigated. The participants were between 22 and 93 years old and received 0–10 rounds of MDA treatment since 2001. The LE stages ranged from 1 to 7 and the CFA+MF+ individuals had a median of 163 MF/mL (range 2–1127 MF/mL). An overview about the characteristics of the study population is shown in Table 1.

### 3.2. Higher Frequencies of IL-10+ Polymorphonuclear MDSCs in Patent W. Bancrofti-Infected Individuals

Based on the gating strategy (Appendix A), the composition of MDSCs was analyzed. In general, MDSCs consist of two distinct sub-populations, monocytic (Mo)-MDSCs and polymorphonuclear (PMN)-MDSCs [20,21,22,23]. The frequencies of total MDSCs (CD11b+CD33+HLA-DR-) (Figure 1A) and the frequencies of PMN-MDSCs (CD11b+CD33+HLA-DR-CD14-CD15+) (Figure 1B) were not significantly different between the groups. Mo-MDSCs (CD11b+CD33+HLA-DR-CD14+CD15-) could not be detected in the peripheral blood of the different cohorts. In regard to cytokine expression, IL-6+PMN-MDSCs could not be detected, but the patent individuals (CFA+MF+) had significantly higher frequencies of IL-10+PMN-MDSCs compared to the latent individuals (CFA+MF-; Figure 1C). Next, in order to ascertain if the status of infection (latent vs. patent) after MDA influences the immune cell composition and function, the PI group was sub-classified according to the MF status detected in 2009 before the intensification of MDA programs. However, the former MF status of the individuals in the PI group had no impact on the PMN-MDSC composition (Appendix A). Taken together, these data show that the patent individuals are characterized by elevated PMN-MDSCs that are additionally positive for the immunosuppressive cytokine IL-10 (IL-10+PMN-MDSCs).

### 3.3. Comparable Frequencies of M2 Macrophage Populations between the Study Cohorts

To investigate if the monocyte/macrophage populations are different between the study cohorts, we further analyzed the frequencies of CD14+ monocytes and M2 macrophages and their cytokine expression according to the applied gating strategy (Appendix A). The frequencies of CD14+ cells were comparable, except that the PI group showed a slight decrease compared to the CFA+MF− individuals (Figure 2A). Since M2 macrophages were shown to play a crucial role in the filarial-driven immune regulation [44,45], we analyzed the frequencies of M2 macrophages (HLA-DR-CD206+CD163+CD200R+) and their cytokine expression. In general, no differences could be observed in the M2 macrophage frequencies between the different cohorts (Figure 2B), and cytokine expression was also comparable between the patient groups (Figure 2C,D). In regard to the prior MF status of the PI group, the frequencies of CD14+ cells in the PI MF- patients were significantly lower compared to the CFA+MF− individuals, but no differences could be observed between the PI MF+ and PI MF− groups (Appendix A). In addition, the levels of M2 macrophages and their IL-4 expression were comparable between the groups (Appendix A), whereas the PI MF+ individuals showed significantly higher frequencies of IL-10+ M2 macrophages compared to the CFA+MF+ individuals, but again, no significant differences were obtained between the PI MF+ and PI MF− cohorts (Appendix A). Overall, these findings show that the frequencies of M2 macrophages are comparable between the patient cohorts and are not influenced by MDA.

### 3.4. Reduced ILC2 Frequencies in W. bancrofti-Infected Individuals

Finally, we analyzed the ILC subsets according to the applied gating strategy (Appendix A). Interestingly, we observed no significant differences between the patient cohorts when considering the total ILCs (Lin-CD127+), Lin-CD127+CD294−CD117−ILC1 and Lin-CD127+CD294−CD117+ILC3 subsets (Figure 3A–C), while the frequencies of ILC2 (Lin-CD127+CD294+) were significantly lower in the *W. bancrofti*-infected individuals compared to the EN and PI individuals (Figure 3D). The cytokine expression patterns in the different ILC subsets were negligible. Further analysis of the PI group according to the MF status revealed no differences between the total ILCs, ILC1 and ILC3 frequencies (Appendix A). Nevertheless, the PI MF− individuals had increased ILC2 frequencies compared to the latent *W. bancrofti*-infected individuals, but no differences could be observed between the PI MF+ and PI MF− groups (Appendix A). These findings show that the ILC2 populations are reduced in the periphery of the *W. bancrofti*-infected individuals and recover after infection clearance due to MDA.

## 4. Discussion

The modulation of the host immune system during filarial infection is a highly complex process that has not yet been fully deciphered. Many previous studies demonstrated that a variety of cell types is involved in this immune modulation, but studies about M2 macrophages, MDSCs and ILCs are scarce, especially during human filarial infections. Here, we show that the frequencies of IL-10+PMN-MDSCs are elevated in the patent *W. bancrofti*-infected individuals, whereas the ILC2 frequencies are reduced. Importantly, our study design not only allowed the comparison of the changes in myeloid cells in different *W. bancrofti* infection groups, but also the evaluation of the impact of MDA. Interestingly, we saw that the clearance of infection results in a recovery of the peripheral ILC2 population comparable to uninfected controls (EN). In line with this, a study reported that the proportions of ILC2s increased after curative anti-helminthic treatment in *Schistosoma haematobium*-infected infants, suggesting that the presence or large numbers of ILC2s hinder helminth development [31]. Nonetheless, the EN patients displayed increased ILC2 levels when compared to the CFA+MF+ individuals. Previous studies showed that ILCs were present in different tissues and peripheral blood during filarial infections [46]. Therefore, it seems that steady amounts of ILC2 are associated with worm/MF-free individuals, or ILC2 levels are restored in the periphery to basal numbers after the disease is cleared. Using the *L sigmodontis* mouse model, it was shown that ILC2 populations expand during infection [47]. In the current study, ILC2 levels were decreased in the periphery of *W. bancrofti*-infected individuals when compared to the EN and PI groups, but there were no statistical differences between the patent and latent groups showing that the presence of adult worms *per se* influences the presence and migration of this particular cell population. A study using cryopreserved PBMCs reported elevated ILC2 levels in a cohort of 21 mixed filarial-infected individuals (*O. volvulus*, *Loa loa*, *W. bancrofti*) when compared to 11 uninfected donors. However, these authors only had three *W. bancrofti*-infected patients in their group and did not differentiate according to the status of the patient (MF or LE status) in their analysis [32]. The present results were derived from a larger sample size. Although we had noticeable differences in the group sizes (e.g., *n* = 14 CFA+MF+ vs. *n* = 117 PI), it shows a clearer picture of the appearance of peripheral ILC2 during a *W. bancrofti* infection. No correlation between the LE stage and ILC2 and the other cell populations could be observed. Of note, it was reported that the expression of markers in ILC2 populations such as CD294 can fluctuate [48]. Thus, the described ILC2 populations need to be referred to as CD294+ ILC2. However, ILCs were shown to differentially migrate through various barrier and non-barrier tissues to initiate tissue-specific responses to pathogens [46,49]. Thus, further investigations need to elucidate if the decreased ILC2 frequencies measured in the periphery in *W. bancrofti*-infected individuals is caused by the migration of these cells to various tissues, especially at the site of infection (lymphatics) [50]. Indeed, it was shown that Tregs can suppress ILC2 function in an allergic airway inflammation model using humanized mice [51]. However, a follow-on study comparing the immune cell frequencies in peripheral blood and lymph fluid with increased sample numbers of patent *W. bancrofti*-infected individuals, which is a difficult task due to intensified MDA programs in Ghana, needs to be performed to clarify this research question. In addition, studies need to investigate in more detail the different ILC subsets by using more markers such as CD45 and NKp44; a study has noted that these markers are only required for the characterization of ILCs in tissue [42], while another study reported that ILC3s are mostly NKp44- [50]. However, it was shown that ILC precursors (ILCp) are predominant in blood and ILC3 likely stem from ILCp [48,52,53,54]. Thus, the ILC3 subset as defined in this work should be considered as a mixed ILCp/ILC3 population. It is well-accepted that ILCs release cytokines crucial for their functionality [55], but we could only measure very low or negligible amounts of cytokines in this study. However, the findings from ex vivo cytokine levels are difficult to interpret due to low/undetectable levels of intracellular cytokines. Therefore, future investigations need to perform in vitro re-stimulation experiments to detect cytokine production from distinct cell populations. In general, the numbers of ILCs were considerably low in peripheral blood independently of the patient group. This might be due to the low amount (100 µL) of whole blood used for the flow cytometry analysis. Additional studies using a higher volume of peripheral blood need to be performed to characterize ILCs in filarial-infected individuals.

In regard to MDSC populations, we could not obtain Mo-MDSCs and did not observe the differences of PMN-MDSCs between the patient groups but revealed increased frequencies of IL-10+PMN-MDSCs in the patent *W. bancrofti*-infected individuals compared to the latent *W. bancrofti*-infected individuals. This contrasts with our data in the murine model that have demonstrated that Mo-MDSCs and PMN-MDSCs infiltrated the site of infection of *L. sigmodontis*-infected BALB/c mice and that only Mo-MDSCs exhibited suppressive capacities on the production of IL-13 and IFN-γ by CD4+ T cells [23]. Again, this conflicting finding might be related to the milieu from which Mo-MDSCs were obtained. For instance, in the murine study, these cells were isolated from the thoracic cavity, the site of infection, and thus, results from the periphery cannot be compared with the results from the murine model of filariasis. Similarly, peripheral M2 macrophages were comparable between the groups, despite the fact that studies revealed that helminth infections induce M2 macrophages, but these findings are based on murine experimental models [56,57]. Nevertheless, it was shown that *Brugia malayi* infection is associated with monocytes that are characterized by an alternatively activated immunoregulatory phenotype, but these results were obtained through in vitro stimulation of monocytes with *B. malayi* antigen or MF [44,58]. These findings suggest that M2 macrophages reside at the site of infection or differentiate upon filarial antigen trigger. As mentioned above, the analysis of lymph fluid samples and in vitro restimulation assays are needed for future studies to obtain insight into the function of macrophage, MDSC and ILC subsets during lymphatic filariasis.

A recent study on a cohort from a region highly endemic for *W. bancrofti* has reported that high levels of IL-10 and Tregs were associated with the absence of MF sheath antibodies [59]. We have shown that the individuals who cleared the infection due to MDA had the lowest IL-10 and Tregs levels when compared to both the latent and patent patients, suggesting that IL-10 might serve as a signature for MF release or ongoing filarial infection. In support of this, we have previously shown that the subjects from the PI group are comparable to the EN individuals having reduced frequencies of Tregs and IL-10-producing Bregs compared to the infected groups [5]. Interestingly, the current study shows that the frequencies of IL-10+ PMN-MDSCs were highest in the patent *W. bancrofti*-infected, indicating that MDSCs are critical for the induction of regulatory immune responses [60], which is a characteristic of patent *W. bancrofti*-infections [5,8]. Therefore, we hypothesize that the reduced ILC2 frequencies might be due to the increased regulatory milieu with increased frequencies of Tregs, Bregs and IL-10-expressing MDSCs.

## 5. Conclusions

We showed that the peripheral frequencies of MDSCs, M2 macrophages, and ILC1 and ILC3 were not influenced by *W. bancrofti* infection in a study cohort from the Western Region of Ghana. However, significantly reduced frequencies of ILC2 in the patent *W. bancrofti*-infected individuals were measured, suggesting that the regulatory milieu, e.g., increased frequencies of Tregs, Bregs and IL-10+ PMN-MDSCs, drive the decreased numbers of ILC2 in the peripheral blood. However, the clearance of infection restored the ILC2 frequencies in the periphery, highlighting that filarial-induced alterations of the peripheral immune composition require an active *W. bancrofti* infection.

## Figures and Tables

**Figure 1 pathogens-12-00665-f001:**
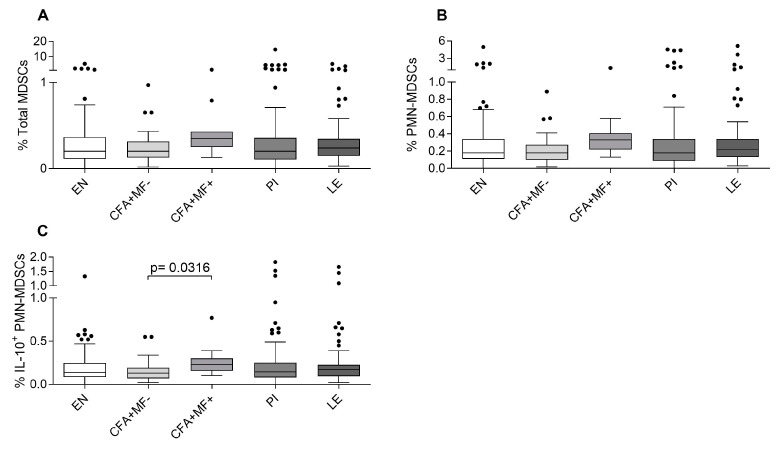
Increased IL-10+PMN-MDSC frequencies in peripheral blood of patent *W. bancrofti*-infected individuals. Using flow cytometry, peripheral whole blood cells from endemic normal (EN; *n* = 89), latent (CFA+MF-; *n* = 54) and patent (CFA+MF+; *n* = 14) *Wuchereria bancrofti*-infected, previously infected individuals (PI; *n* = 117) and individuals who suffer from lymphoedema (LE; *n* = 85) were analyzed for frequencies (%) of (**A**) total MDSCs (CD11b+CD33+HLA-DR-), (**B**) PMN-MDSCs (CD11b+CD33+HLA-DR-CD14-CD15+) and PMN-MDSC expressing (**C**) IL-10. Graphs show box whiskers with median, interquartile ranges and outliers (dots). Statistical significances between the indicated groups were obtained after a Kruskal–Wallis test followed by a Dunn’s multiple comparison post hoc analysis.

**Figure 2 pathogens-12-00665-f002:**
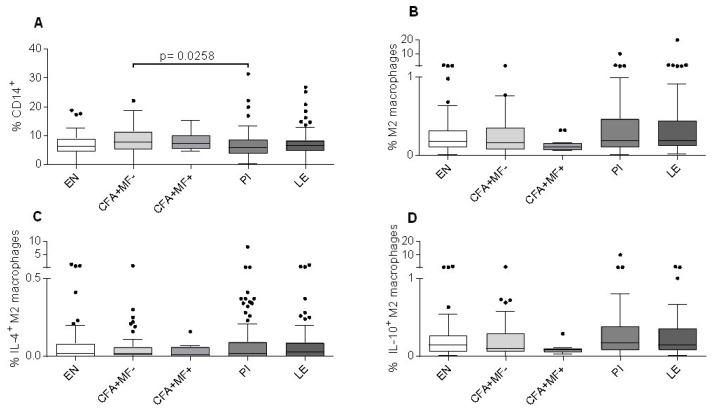
Comparable frequencies of M2 macrophages in peripheral blood of the study cohort. Using flow cytometry, peripheral whole blood cells from endemic normal (EN; *n* = 89), latent (CFA+MF−; *n* = 54) and patent (CFA+MF+; *n* = 14) *Wuchereria bancrofti*-infected, previously infected individuals (PI; *n* = 117) and individuals who suffer from lymphoedema (LE; *n* = 85) were analyzed for frequencies (%) of (**A**) CD14+ cells, (**B**) M2 macrophages (HLA-DR-CD206+CD163+CD200R+) and (**C**) M2 macrophages expressing IL-4 or (**D**) IL-10. Graphs show box whiskers with median, interquartile ranges and outliers (dots). Statistical significances between the indicated groups were obtained after a Kruskal–Wallis test followed by a Dunn ‘s multiple comparison post hoc analysis.

**Figure 3 pathogens-12-00665-f003:**
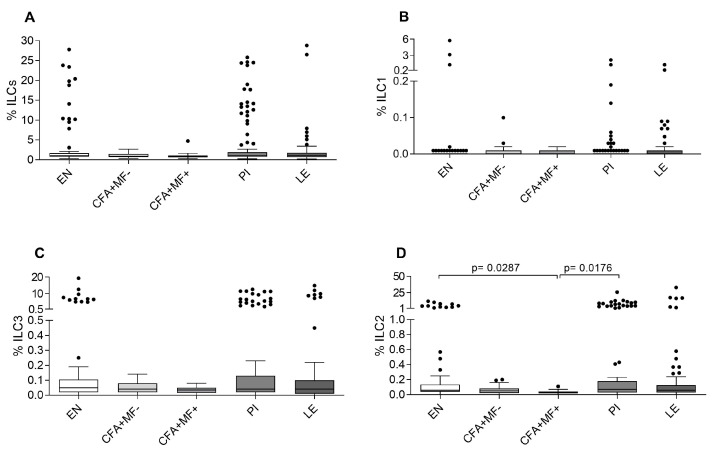
Reduced frequencies of ILC2 in *W. bancrofti*-infected individuals. Using flow cytometry, peripheral whole blood cells from endemic normal (EN; *n* = 89), latent (CFA+MF−; *n* = 54) and patent (CFA+MF+; *n* = 14) *Wuchereria bancrofti*-infected as well as previously infected individuals (PI; *n* = 117) and individuals who suffer from lymphoedema (LE; *n* = 85) were analyzed for frequencies (%) of (**A**) total ILC (Lin-CD127+SSC−), (**B**) ILC1 (Lin-CD127+SSC-CD294−CD117−), (**C**) ILC3 (Lin-CD127+SSC-CD294-CD117+) and (**D**) ILC2 (Lin-CD127+SSC-CD294+). Graphs show box whiskers with median, interquartile ranges and outliers (dots). Statistical significances between the indicated groups were obtained after a Kruskal–Wallis test followed by a Dunn ‘s multiple comparison post hoc analysis.

**Table 1 pathogens-12-00665-t001:** Characteristics of study population. Overview about number, age, gender, rounds of MDA since 2009, microfilariae (MF) counts and lymphoedema (LE) stages of endemic normal (EN), latent (CFA+MF−) or patent (CFA+MF+) *W. bancrofti*-infected as well as previously infected individuals (PI) who cleared the infection due to MDA programs and individuals who suffer from LE.

	EN (*n* = 89)	CFA+MF− (*n* = 54)	CFA+MF+ (*n* = 14)	PI (*n* = 117)	LE (*n* = 85)
Gender [Female/Male]	75/14	20/34	1/13	56/61	64/21
Percentage gender [Female/Male]	84.27/15.73	37.04/62.96	7.14/92.86	47.86/52.14	75.29/24.71
Median age years (Range)	46 (27–80)	45 (25–64)	37 (22–58)	40 (23–93)	52 (17–70)
Median MDA rounds (Range)	5 (0–10)	3 (0–10)	1 (0–6)	3 (1–10)	5 (0–10)
Median LE stage (Range)	-	-	-	-	3 (1–7)
Median microfilaria count [MF/mL] (Range)	-	-	163 (2–1127)	-	-

## Data Availability

The data presented in this study are available on request from the corresponding author. The data are not publicly available due to ethical restrictions since they contain information that could compromise the privacy of research participants.

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
