# Peer review of "Reduced Type 2 Innate Lymphocyte Cell Frequencies in Patent Wuchereria bancrofti-Infected Individuals"

_pathogens, 2023, doi:10.3390/pathogens12050665_

Round 1
Reviewer 1 Report
The authors provide an overview of LF, a mosquito-borne parasitic infection caused by W. bancrofti. Here, they explore the immune responses associated with W. bancrofti infection. Overall, the article provides useful information on LF, the immune responses associated with the infection, and the potential role of innate immune cells in the disease. The study shows that in patent W. bancrofti-infected individuals, the frequencies of IL-10+PMN-MDSCs are elevated, whereas ILC2 frequencies are reduced. The study also shows that anti-filarial treatment results in the recovery of the peripheral ILC2 population comparable to uninfected controls. The study discusses the differences in ILC subsets and their frequencies in various groups, such as EN, patent, latent, and leg lymphedema (LE) groups. The authors suggest that the decreased ILC2 frequencies observed in W. bancrofti-infected individuals may be caused by their migration to various tissues, especially the site of infection (lymphatics). The authors conclude by noting that further studies are needed to clarify the research question, including studies with increased sample numbers from patent W. bancrofti-infected individuals.
I have no additional comments.
Reviewer 2 Report
The study from Tamadaho, Arndts, Osei-Mensah et al. outlines the frequencies of ILCs, MDSCs and M2 macrophages in peripheral blood of individuals from Ghana living in areas endemic for the filarial parasite Wuchereria bancrofti. It is a strength of this manuscript that large cohorts were available for these studies allowing to compare individuals which are resistant to infection, recently recovered from infection due to treatment or currently acutely of chronically infected. This allows valuable analyses to correlate these groups of people to cell frequencies observed. Up to this stage this information has relevance to establish biomarkers. However, I cannot endorse publication of this manuscript straight away, because there are a few points the authors need to address before of that.
Specific points:
1. For the ILC gating strategy a Lin vs CD127 plot should be used as the populations used are continous and no FMO is provided.
2. Show the FMO for CD294 and IL-10
3. The population annotated as ILC3 is actually a mixed ILCp/ILC3 population and this should be reflected in the manuscript. If in doubt, see recent publications from the Di Santo and Mjosberg groups. Also call the ILC2 CD294+ ILC2 because the expression of CD294 in ILC2 is dynamic and human ILC2 can also loose its expression.
4. Replace the term "lymphedema" with "lymphoedema" across the manuscript.
5. Line 33: write "uninfected"
6. Line 54-57: rephrase
7. The introduction should outline what is known about frequencies of T cell subsets, B cells, NK cells and granulocytes in the blood of the groups of people analysed in this study. This should at least address W. bancrofti but could also extend to Brugia spp, Loa loa, Onchocerca volvulus etc.
8. Line 66- 69: rephrase and break up in 2 sentences
9. Line 69: Is this in vitro?
10. Throughout the introduction it needs to be clear whether cells in the blood, tissue, lymph or peritoneal cavity are meant.
11. Across the manuscript it needs to be clarified what frequency means. Perhaps % of CD45+ cells?
12. Line 77: How were ILC depleted?
13. Line 78: What does the release of Mf indicate?
14. Line 78: remove "furthermore" as this sentence contradicts the previous one.
15. which IL-5-producing cells are meant?
16. Line 84:delete: "However,......,thus," and replace with "In order to analyse .... in human blood....."
17. Line 107: explain why you took the blood at night. At what time of the day are Mf present in the peripheral blood? Add reference such as one of the old Hawking studies.
18. Is the CFA test specific for W. bancrofti? Clarify in the methods.
19. Line 118 and 121: I believe you used ml for these facs analyses....
20. Line 180: could not be
21. Figure 1: c is invisible and replace a and b with Sfigure 2.
22. Line 210: delete furthermore
23. Figure 2: replace with Sfigure 3
24. Figure 3: replace with Sfigure 4, can you correleate ILC2 frequencies to LE stages?
25. For all figures: show key facs plots to substantiate the data.
26. Line 284: The cells do not expand, but the cell population does
27. Line289-293: rephrase
28. Line 305: Indeed, it has
29. Line 336: helminth (without the s)
30. Line 355: delete regulatory
31. Line 355: for the induction
Reviewer 3 Report
In this paper investigators Tamadaho et al., have investigated the composition of myeloid-derived suppressor cells (MDSCs), macrophage subsets, and innate lymphoid cells (ILCs), in an active filarial infection Vs latent W. bancrofti-infected individuals. In this study, they enlighten that the filarial-driven changes of the immune responses require an active infection and are not maintained upon clearance of infection. The parameters in the study are properly evaluated and the manuscript is well written. Methods and results are clearly discussed in the manuscript. While this information is generally of interest and very educative, there are several points that have to be addressed by the authors.
Suggestions/minor comments:
- In the abstract, line no. 21, authors have mentioned “the treatment has led to……it would be good if they mention which treatment (e.g MDA, etc.)
- In the introduction, it would be good to mention what type of treatments have been implemented to reduce the number of infected individuals.
- Line no, 122-124, there is discontinuity, (‘remaining’ cells were then fixed, and… this is confusing) please rephrase the entire paragraph accordingly.
- In most of the figures, authors have shown the data in percent changes (e.g % total MDSCs, % M2 macrophages, etc.), however, it is not clear whether the data shown is MEAN +/- ?. Although it has been shown in % changes, I could see the error bars, what does that represent? SD or SEM?
- The quality of the figures is poor, please replace all with high resolution.
- There are a lot of definite and indefinite articles missing along with some grammatical errors, also please correct the manuscript for undefined spaces, etc.
Reviewer 4 Report
This study investigated the profile of modified Th2 type cellular responses in the peripheral blood of individuals with various status of W. bancrofti and post MDA. Although the sample size is quite large, but the number of mf+/CFA+ is very few, still there are interesting findings. The results and discussion were nicely written. There are few grammatical errors which need to be corrected.
Questions:
1 1. As the aim of the study was to investigate composition of MDSCs, M2 macrophages and ILCs, how the authors could conclude that significantly reduced frequencies of ILC2 in patent W. bancrofti-infected individuals suggesting the regulatory milieu, e.g., increased frequencies of Treg, Breg and IL-10+ PMN-MDSCs, drive the decreased numbers of ILC2 in the peripheral blood. Did the authors also analyze the Treg and Breg ?
2 2. MDA rounds was lowest in the patent infection group, would it affect the analyses of post infection?
Round 2
Reviewer 2 Report
The authors provided a much improved version of the manuscript, but I still need 2 questions to be re-addressed.
Specific Issues
1. The authors should backgate all ILC subsets as Lin vs CD127 to demonstrate there are no contaminating cells. If available, FMOs for Lin and CD127 would be helpful.
2. Line 151 is missing from the manuscript, and I cannot see how the authors answered former question 11. The authors appear to calculate the frequencies using a total cell population that contains hematopoietic cells, RBC and platelets. Do the authors think this may be an issue?
Round 3
Reviewer 2 Report
Thank you for providing the FMOs in SFig1.
The staining for lineage markers appears very dim, and it is still not convincing that you have gated a pure ILC population. Hence, it is still pivotal to provide a Lin vs CD127 plot as this should help to separate out the ILC population from which you analysed all ILC subsets. You may have to manipulate the axes to increase the separation from Lin-positive cells.
